# Seasonal controls override forest harvesting effects on the composition of dissolved organic matter mobilized from boreal forest soil organic horizons

Keri L. Bowering[1], Kate A. Edwards[2], Susan E. Ziegler[1]

[1]Department of Earth Sciences, Memorial University of Newfoundland, St. John's, A1C 5S7, Canada
[2]Natural Resources Canada, Canadian Forest Service, Ottawa, K1A 0E4, Canada

*Correspondence to*: Susan E. Ziegler (sziegler@mun.ca)

**Abstract.** Dissolved organic matter (DOM) mobilized from the organic (O) horizons of forest soils is a temporally dynamic flux of carbon (C) and nutrients, and the fate of this DOM in downstream pools is dependent on the rate and pathways of water flow as well as its chemical composition. Here, we present observations of the composition of DOM mobilized weekly to monthly from O horizons in mature forest and adjacent harvested treatment plots. The study site was experimentally harvested, without replanting, 10-years prior to this study. Thus, the treatments differ significantly in terms of forest stand and soil properties, and interact differently with the regional hydrometeorological conditions. This presented an opportunity to investigate the role of forest structure relative to environmental variation on soil DOM mobilization. On an annual basis, fluxes of total dissolved nitrogen (TDN) and dissolved organic nitrogen (DON) were largest from the warmer and thinner O horizons of the harvested (H) treatment compared to the forest (F) treatment, however, neither phosphate or ammonium fluxes differed by treatment type. On a short-term basis in both H and F treatments, all fluxes were positively correlated to water input, and all concentrations were positively correlated to soil temperature and negatively correlated to water input. Soil moisture was negatively correlated to the C:N of DOM. These results suggest common seasonal controls on DOM mobilization regardless of harvesting treatment. Optical characterization of seasonally representative samples additionally supported a stronger control of season over harvesting. The chemical character of DOM mobilized during winter and snowmelt: lower C:N, higher specific ultraviolet absorbance and lower molecular weight of chromophoric DOM (CDOM; higher spectral slope ratio), was representative of relatively more decomposed DOM, compared to that mobilized in summer and autumn. This shows that the decomposition of soil organic matter underneath a consistently deep snowpack is a key determinant of the composition of DOM mobilized from O horizons during

winter and the hydrologically significant snowmelt period regardless of harvesting impact. Despite the higher
proportion of aromatic DOM in the snowmelt samples, its lower molecular weight and rapid delivery from O to
mineral horizons suggests that the snowmelt period is not likely to be a significant period of DOM sequestration
by mineral soil. Rather, the higher molecular weight, high C:N DOM mobilized during slow and relatively
infrequent delivery during summer and rapid, frequent delivery during autumn are more likely to support
periods of mineral soil sequestration and increased export of fresher terrestrial DOM, respectively. These
observed seasonal dynamics in O horizon DOM suggest the predicted decreases winter and spring snowfall and
increasing autumn and winter rainfall with climate warming in this region will enhance mobilization of DOM
that is more reactive to mineral interactions in deeper soil, but also more biological and photoreactive in the
aquatic environment. Understanding the downstream consequences of this mobilized DOM in response to these
shifts in precipitation timing and form can improve our ability to predict and manage forest C balance but
requires understanding the response of landscape hydrology to these changing precipitation regimes.



## 1 Introduction

Dissolved organic matter (DOM) mobilized from organic horizons of forest soils represents an ecologically significant source of carbon (C) and nutrients both within forest catchments (Qualls and Haines, 1991), and from soils to aquatic systems (Jansen et al., 2014). The fate of mobilized soil DOM is influenced by both water flow dynamics (rate and pathways) and the chemical composition of DOM (Roulet and Moore, 2006), although the interaction of these two factors is not often captured in soil studies. The composition of mobilized organic horizon DOM is the net result of production and uptake processes, as well as the relative solubility of organic matter inputs from different sources. While soil extractions provide valuable information on potential sources, bioavailability, and production mechanisms of soil DOM (i.e. Jones and Kielland, 2012; Hensgens et al., 2020), as well as transformation and fate in mineral soil (Kothawala et al., 2008), they cannot capture the interaction of these factors with local hydrometeorological conditions important to understanding the net movement of DOM *in situ*. Measurements that incorporate the role of soil hydrology with DOM mobilization place the knowledge gained from extraction studies into the larger catchment scale context. For instance, these measurements can more directly inform regional chromatography (i.e. Kaiser and Kalbitz, 2012; Shen et al., 2014) and terrestrial-to-aquatic C flux conceptual models (i.e. Tank et al., 2018), which further allow us to assess the impacts of disturbances such as harvesting and climate change on landscape C balance (Casas-Ruiz et al. 2023).

Forest C and nitrogen (N) cycles are tightly linked and the C:N of bulk soil provides clues about ecosystem functioning and the bioavailability of soil organic matter. Similarly, the C:N of DOM is considered a measure of DOM bioavailability (McDowell and Likens, 1988; McDowell et al., 2004; McGroddy et al., 2008). However, while C:N of DOM correlates to C:N of soil in some studies (Gödde et al., 1996; Michalzik and Matzner, 1999), it does not in others (Cortina et al., 1995; Michel and Matzner, 1999). The mobilization of dissolved organic C (DOC) relative to dissolved organic N (DON) is correlated on an annual basis (Michalzik et al., 2001), but whether this correlation holds across seasons is not known and could help explain the discrepancies in the relationship between soil C:N and DOM C:N. Additionally, boreal forests accumulate particularly large amounts of C in surface layers because of temperature-limitations on soil organic matter decomposition and the recalcitrance of coniferous tree litter and forest floor mosses (Philben et al. 2018; Hensgens et al., 2020). Nitrogen-limitations can also affect decomposition and soil C accumulation (Averill and Waring, 2018), and may explain why the C:N of boreal soil organic horizons is higher in areas not affected by industrial N deposition (for instance, Alaskan compared to Swedish boreal forests), with likely effects on soil DOM. These

dynamics are further impacted by snow-cover, especially during seasonal transition periods (Groffman et al., 2018), but the type of snowpack change is regionally variable with differing effects on the underlying soil (Stark et al., 2020).

Spectroscopic characterization (absorbance and fluorescence) of chromophoric DOM (CDOM) is an efficient technique for describing broad DOM compositional differences in surface waters and identifying terrestrial

DOM sources (Helms et al., 2008; McKnight et al., 2001; Jaffé et al., 2008; Berggren and Giorgio, 2015). These techniques have also been used to assess the compositional variability of terrestrial DOM. In litter incubation experiments, for instance, specific ultraviolet absorbance (SUVA) of leached spruce needles increased during decomposition because of increased solubility of lignin as it became more degraded (Hansson et al., 2010; Klotzbücher et al., 2013). Similarly, an increase in aromaticity of soil DOM, but a decrease in C:N of DOM,

was observed in snowmelt simulation performed over soil columns collected from both coniferous and deciduous sites (Campbell et al., 2014), and the aromatic content of O horizon DOM from different forest types in Sweden were found not to differ (Fröberg et al., 2011). Variations in aromaticity of O horizon DOM suggests variation in mineral stabilization potential of DOM in mineral soils. For instance, an input of aromatic compounds from O horizons to the mineral soil is more likely to result in formation of organo-mineral

complexes that stabilize OM, than an input of mobile, hydrophilic compounds such as carbohydrates (Guggenberger and Zech, 1993; Kaiser and Kalbitz, 2012). However, *in situ*, the consequences of these transformations are dependent on the rate and pathways of water and DOM composition and flow. Clarification is needed on the controls over both the rate and composition of mobilized DOM which could then inform potential mineral stabilization versus export potential of soil DOM from upland forests under varying

hydrometeorological conditions.

The large and small-scale effects of hydrometeorological conditions on DOM dynamics can be confounded by effects of disturbances, such as fire, insects and forest harvesting. Forest harvesting is a significant anthropogenic disturbance in boreal forests with known impacts on soil moisture and temperature during the growing season and increased export of water and DOM (Kreutzweiser et al., 2008). Mid- to long-term effects

on soils in naturally regenerating forest are not well-known, but are likely significant in the situation when clear-cut, or otherwise disturbed, boreal black spruce forests remain open for long periods of time. As seasonal and decadal scale harvesting effects on DOM dynamics are both potentially significant but confounding, our objective with this study was to parse out the main effects of each. Mobilized soil DOM was sampled on a

weekly to monthly basis over a year using a passive pan lysimeters in open harvested plots compared to adjacent mature black spruce forest plots as part of a case study. Previously, we demonstrated that the thickness of the organic horizon was reduced by almost 50% in the open harvested plots, that the quantity of DOC mobilized in the harvested plots was larger than in the forested plots, and that the relationship between water fluxes and mobilized DOC varied seasonally (Bowering et al., 2020). Here, we describe the temporal and spatial variability of DOM composition mobilized from soil organic horizons to better understand the controls of forest stand and soil structure relative to short-term hydrometeorological variability. Focusing on the hot spots and moments of boreal forest DOM mobilization likely sensitive to climate change, these results help identify the "top-down" controls on C and nutrient storage in boreal forest mineral soils and potential fate of forest soil DOM exported to aquatic systems.

## 2 Materials and Methods

### 2.1 Site Description

This study was conducted in an experimental forest at the Pynn's Brook Experimental Watershed Area (PBEWA) located near Deer Lake, western Newfoundland and Labrador, Canada. (48° 53'14" N, 63° 24' 8" W). The forest is mesic relative to other areas of the boreal ecozone, located in the Maritime Low Boreal Ecoclimate (Lbm) of the Ecoregions Working Group (1989), and is dominated by black spruce (*Picea mariana*). The site is underlain by humo-ferric podzolic soils (Soils Classification Working Group, 1998) that have developed on fresh glacial, poorly sorted till deposits of granite, porphyry, sandstone, and siltstone clasts (Batterson and Catto 2001) with quartz most abundant, followed by plagioclase, muscovite, chlorite, K-feldspar and biotite (Patrick et al., 2022). The surrounding boreal landscape is characterized by a mosaic of different age classes resulting from a history of periodic disturbance, including that from forest harvesting. The climate of the study region is characterized by winters with consistent snowpack for 4-5 months of the year from December to April. The area receives on average 1095 mm of precipitation annually, with approximately 40% as snow, with a mean annual temperature of 3.6°C (Environment Canada Climate Normals, Deer Lake Airport 1981–2010). During the study year total precipitation was 1402 mm with 516 mm (37%) of that as snowfall water equivalent, and snowpack at the start of snowmelt period during the study ranged from 83 to 110 mm snowfall water equivalent across the study plots, and was higher in the harvested plots (Bowering et al., 2020) similar to the long-term average in the region (Environment Canada Climate Normals, Deer Lake Airport 1981–2010). The

site consists of 2 hectares divided into eight 50 x 50 m plots (Fig. S1). Four of the plots were left un-harvested and four were randomly selected for clear-cut harvesting. The four clear-cut plots were harvested on July 07–10, 2003 using a short-wood mechanical harvester, with minimal disturbance to the underlying soil and with any deciduous trees left standing. The close proximity of the plots enabled comparison of mature and harvested forest areas over a similar slope, aspect, elevation and soil type and thus represents a case study of the harvesting effects within the context of seasonal variation. The two treatments of this case study will be referred to as harvested (H) and mature forest (F) throughout. In this study we used three plots of each treatment as indicated in Fig. S1.

Following common forestry practices for the area, the harvested plots were not replanted following clear-cutting. Moss coverage persisted in the H plots where a larger proportion was *Sphagnum sp*. as compared with F plots where moss cover was dominated by *Hylocomium splendens*. The harvested plots also consisted of naturally regenerated herbs (including *Cornus canadensis*, *Chamerion angustifolium*, *Vaccinium angustifolium*) and shrubs (*Alnus alnobetula*) and few young conifers (ages 5-8 years; below 1.5 m in height) at much lower density than the adjacent mature stands (see Fig. S1 for aerial image taken year after lysimeter installation). Average (±SD) organic horizon thickness was 8.2 ± 0.6 and 4.3 ± 0.6 cm, and C stocks were 2.4 ± 0.2 and 1.3 ± 0.3 kg C m$^{-2}$ in the F and H plots, respectively (Bowering et al., 2020). Further information on site preparation and conditions can be found in Moroni et al., (2009) while further soil features (e.g. hydrologic properties) can be found in Bowering et al. (2020).

**2.2 Sampling Design, Lysimeter Installation and Sample Collection**

Passive pan lysimeters were installed just underneath the organic horizon. Each lysimeter has a 0.12 m$^2$ footprint and collects water percolating through the O horizon with a maximum solution collection capacity of 25 L. The lysimeters were designed using reported recommendations for achieving accurate volumetric measurements of soil leachate (Radulovich and Sollins, 1987; Titus et al., 2000). A detailed description of the lysimeter design can be found in Bowering et al. (2020) and illustration of the design and installation of the lysimeters is provided in Fig. S2. Installation of lysimeters began in July 2012 and was completed the following spring in May 2013. The design includes the fixed effect of stand type (mature and harvested forest; F and H, respectively) at a scale above the individual lysimeters. Four lysimeters were installed in each of the three plots of each treatment for a total of 12 F plot lysimeters and 12 H plot lysimeters (Fig. S1). Collection began in July 2013 and sampling of all lysimeters in both treatments (n = 24) was carried out on a weekly to monthly basis,

with the exception of the winter season when only one collection was made, for a total of 27 sampling days within the study year. Lysimeter samples were stored in a cooler immediately following collection. Once transported back to the laboratory the pH of each sample was measured, and then samples were filtered using pre-combusted GF/F (0.45 μm pore size) Whatman filter paper, preserved with mercuric chloride within 24 hours of collection, and stored at 4°C in the dark until analysis. This approach, capturing both vertical and lateral flow, previously revealed an effect of stand type on the mobilization of DOC with the H plots exhibiting a nearly 50% increase in DOC mobilization relative to the F plots (Bowering et al. 2022).

**2.3 Environmental Monitoring**

Three soil temperature and moisture probes per treatment (Decagon ECH2O-TM) were installed mid-organic horizon at approximately 5 cm depth, and two were installed in the mineral layer at approximately 15 cm depth. Soil moisture was measured as % volumetric water content (VWC). One tipping bucket rain gauge (RST Instruments Model TR-525) was installed in an open area on site to monitor local rain and air temperature. Data from this tipping bucket were compared with regional rainfall and air temperature (T) reported by Environment Canada at the Deer Lake Airport (49°13'00" N, 57°24'00" W) approximately 50 km away, and Deer Lake Airport data was a good predictor of the PBEWA rainfall and air T on a weekly basis ($R^2 = 0.882$, p<0.0001). Regional data from the Deer Lake Airport were used to fill a gap in our onsite daily rainfall and mean daily air temperature data between July 7th and 24th, 2013. Snowmelt water input was estimated using changes in snow depth between each lysimeter collection day measured near each lysimeter in both H and F. The average snow depth change by treatment was multiplied by an estimated maritime snow density of 0.343 g cm$^{-3}$ (Sturm et al., 2010) to provide an estimated snowmelt water input value. Snowmelt water input estimates were combined with rain (in H) or throughfall (in F) where applicable to give a total water input to the O horizon over each collection period, different from the soil water fluxes independently measured by the lysimeters. Observations of significant lateral flow along the O to mineral horizon interface have been previously reported (Bowering et al., 2020).

**2.4 Chemical Analysis and Flux Calculations**

The DOC and TDN concentration of each lysimeter sample collected was measured using a high-temperature combustion analyzer (Shimadzu TOC-V and TN analyzer, Japan). Nitrate ($NO_3^-$; detection limit = 0.01 mg N L$^{-1}$), ammonium ($NH_4^+$; detection limit = 0.004 mg N L$^{-1}$) and phosphate ($PO_4^{3-}$; detection limit = 0.01 mg P L$^{-1}$)

were measured using QuickChem Methods No. 10-107-04-1-B, 10-107-06-2-A and 10-115-01-1-A respectively, using flow injection analysis (Lachat QuickChem 8500 Series 2, USA). No $NO_3^-$ was detected using this colorimetric method. Total dissolved phosphorus (TDP; detection limit = 2.9 µg $L^{-1}$), aluminum (Al; detection limit = 1.1 µg $L^{-1}$) and iron (Fe; detection limit = 0.3 µg $L^{-1}$) were measured using Inductively Coupled Plasma – Optical Emission Spectroscopy (Perkin Elmer 5300 DV, USA). These measured concentrations, along with the total volume collected by lysimeters, the number of collection days, and the lysimeter collection area were used to calculate a flux (g solute $m^{-2}$ $d^{-1}$). Measures of Fe and Al and their ratio to DOC were included to track reactive metals relevant in these organic soils because of their role in formation of organo-mineral complexes (TORN et al., 1997) as well as their translocation into O horizons from the lower mineral horizon via fungal activity (Clarholm and Skyllberg, 2013).

## 2.5 Seasonal Designations

Lysimeter collections were grouped into four distinct hydrological periods throughout the year described by observed soil moisture, precipitation, soil temperature patterns, together with water flux dynamics measured via the lysimeters as shown in Bowering et al., 2020. Briefly, summer is characterized by fluctuations in soil drying and rewetting, and frequent periods without any O horizon water fluxes. The transition to autumn is described by more consistent soil moisture and frequent precipitation events that resulted in frequent soil water fluxes as temperatures dropped. Winter was characterized by a consistent snowpack that insulated the soil, where soil temperatures were maintained above 0°C despite sub-zero atmospheric temperatures. A very short-term melting event resulted in only a small delivery of water to the soil and therefore, only small cumulative water flux throughout the whole winter. The snowmelt period is characterized by rapid water input to soil, wet soils and increasing soil temperatures.

## 2.6 Absorbance Properties

For optical property measures, a subset of lysimeter samples from F and H treatments was selected to broadly represent the four seasons (Table 5). Each sample was diluted to approximately 15 mg DOC $L^{-1}$ for absorbance measurements. An absorbance scan from 200–800 nm was performed on each sample in a 1 cm cuvette using a Perkin Elmer Lambda UV/Vis spectrophotometer following a blank consisting of NanoUV water (Barnsted Inc). Specific UV absorbance was calculated using the sample absorbance at 254 nm normalized to DOC concentration (SUVA$_{254nm}$). Spectral slopes of the 275–295 nm low molecular weight (LMW) region and 350–400 nm high molecular weight (HMW) region were calculated from each absorbance spectra based on Helms et

al. (2008), and a slope ratio ($S_R$) indicative of the LMW:HMW of CDOM was used to describe changes in relative molecular weight of CDOM. The absorbance spectra were corrected for potential Fe(III) interference, using correction factors based on (Poulin et al., 2014), but derived for these specific sample types. Although the specific speciation of Fe was not measured, a 100% Fe (III) was assumed to facilitate an estimate of the highest

possible interference given the oxic nature of samples when analyzed in the laboratory. A negligible effect of Fe on the absorbance measurement was observed for these samples (Fe represented 0.4 – 0.6% of total sample absorbance per collection date). Seasonally representative absorbance properties ($SUVA_{254nm}$, $SS_{275-295}$, $SS_{350-400}$, and $S_R$), the C:N of DOM, and DOC:Fe were included in a principal component analysis to explore the predominant variables contributing to the effect of treatment relative to season.

**2.7 Statistical Analysis**

A repeated measures linear mixed effects models (RM-LMM) were used to assess the effects of collection day, and the interaction between sampling date and treatment on the fluxes and concentrations using the 'nlme'

package (Pinheiro et al., 2022). Post-hoc Tukey tests using the 'lsmeans' package (Lenth, 2016) were used to determine significant differences between H and F treatments on individual collection days. Further RM-LMMs were used to assess the effects of season and treatment on DOM fluxes and to assess the effects of collection day and treatment on absorbance properties, metals and their ratio to DOC. Diagnostic autocorrelation plots were generated using the ACF function in R (Mangiafico 2016) to assess autocorrelation associated with

lysimeter measurements. These plots demonstrate that each lag point is smaller than the proceeding lag point signifying a lack of autocorrelation. One-way ANOVAs were used to determine differences in total annual fluxes and mean concentrations between H and F treatments over the entire study period. Annual fluxes by treatment are shown in boxplots. Correlation testing was used to examine the association between weekly to monthly lysimeter captured fluxes and concentrations, and environmental predictor variables: mean soil

temperature, mean soil moisture and total water input. Multiple regressions were not used due to the multi-collinearity of the predictor variables, which affected the estimated regression parameters (Quinn and Keough, 2002). Individual Pearson correlations, however, were used here to evaluate the degree of association between variables within the dataset. A Bonferroni correction was applied in the evaluation of these correlations to reduce the type I error. A principal component analysis was performed using the ggfortify (Tang et al., 2016)

and factoextra (Kassambara and Mundt, 2017). All statistical analyses were performed using RStudio version 1.0.136.

## 3 Results

### 3.1 Annual fluxes and concentrations in harvested and forest treatments

Total annual flux of TDN and DON was largest from the organic (O) horizons of the H compared to F treatment
(Fig. 1d,e and Table S1), consistent with DOC, and total water (soil solution) fluxes previously reported
(Bowering et al., 2020). In both treatments DON comprised approximately 85% of the annually mobilized total
dissolved nitrogen (TDN) flux. Ammonium ($NH_4^+$) was the predominant form of inorganic nitrogen (N), with
no detectable nitrate. The annual $NH_4^+$ flux and the annual average C:N of DOM were not different between H
and F treatments. However, the C:N of DOM in the H treatment was higher than the C:N of the O horizon soil
from which it was derived, while the C:N of DOM in the F treatment was similar to the C:N of the O horizon
soil (Fig. 2b,c). Despite treatment differences in annual fluxes of TDN and DON (Fig. 1 and Table S1), the
average annual concentrations of all solutes did not differ between treatments (Table S2).

### 3.2 Intra-annual fluxes and concentrations in harvested and forest treatments

The intra-annual fluxes and concentrations of $PO_4^{3-}$, TDN, $NH_4^+$, DON as well as $NH_4^+$:TDN and C:N of DOM
were variable on a weekly to monthly basis (Fig. 1 and Fig. 2), indicated by the significant effect of collection
day ($p<0.001$; Table 1; Table S3). There was also an interactive effect of treatment and day on all
concentrations and fluxes (Table 1, Fig. 1,and Fig. 2) though the effect of day exhibited p-values much lower
than the interaction in the case of TDN fluxes, DON fluxes and C:N of DOM. An effect of treatment was
detected for DON fluxes only ($p = 0.0208$) with H often exceeding F treatment fluxes. All concentrations were
positively correlated to soil temperature (Table 2a), except for DON in the F treatment, and all concentrations
were

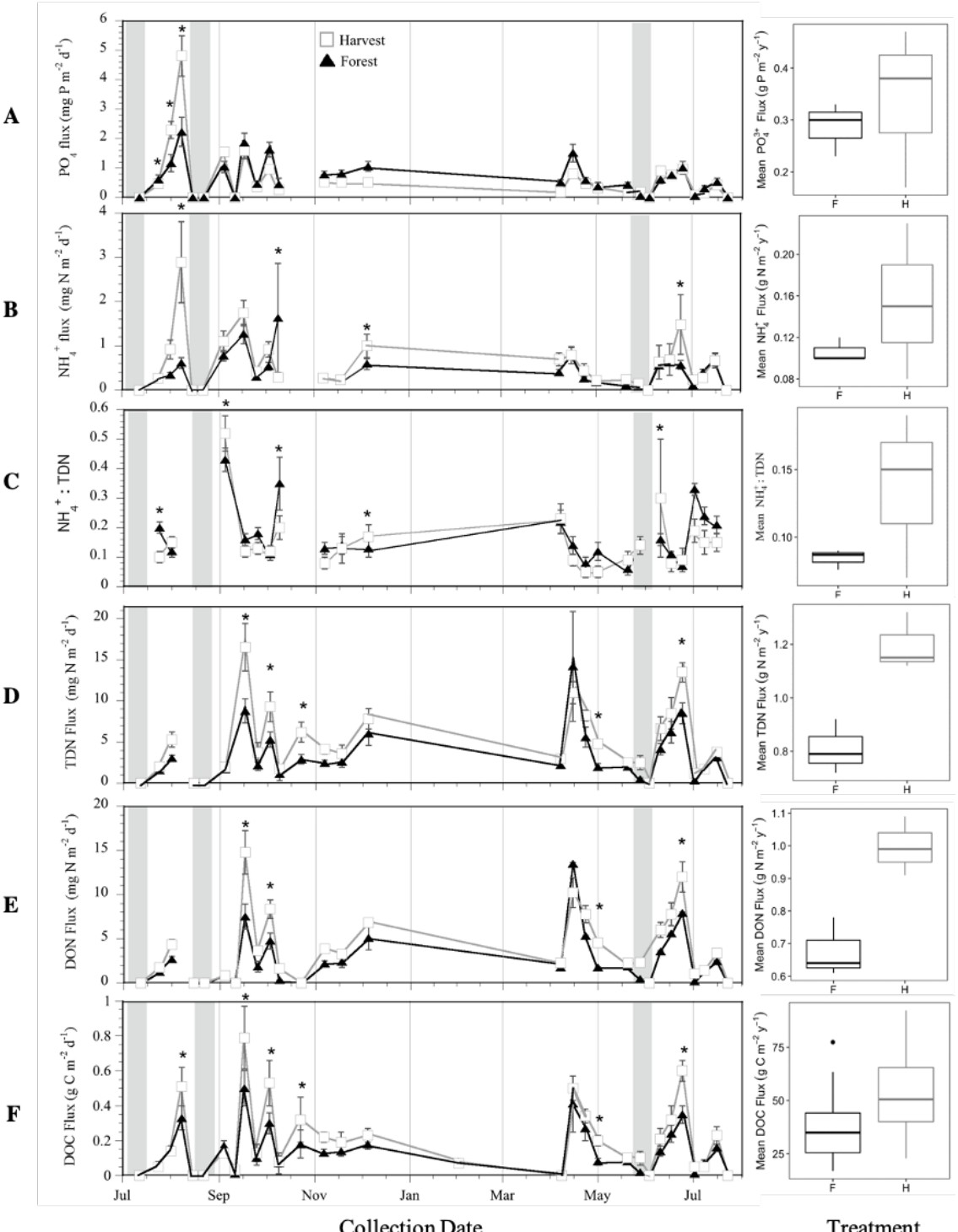

**Figure 1.** Intra-annual variation of lysimeter captured (a) phosphate ($PO_4^{3-}$) fluxes (b) ammonium ($NH_4^+$) fluxes and (c) total dissolved nitrogen (TDN) fluxes, (d) ammonium to total dissolved nitrogen ratio ($NH_4^+$/TDN), and (e) Dissolved organic nitrogen (DON) from soil organic horizons in mature forest (F) and harvested (H) treatment plots located within the Pynn's Brook Experimental Forest in western Newfoundland, Canada. Values are means of 12 lysimeters per treatment. Asterisks indicate significant differences between treatments

determined by repeated measures two-way ANOVA and post-hoc least-square means tests, alpha = 0.05. Grey bars indicate soil drying periods characterized by 10 or more consecutive days receiving less than 10 mm of rainfall. Boxplots show the median (line inside box), upper and lower quartiles (box top and bottom, respectively), and minimum and maximum values (error bars) associated with plot scale annual means (n=3 per plot type; a-e).





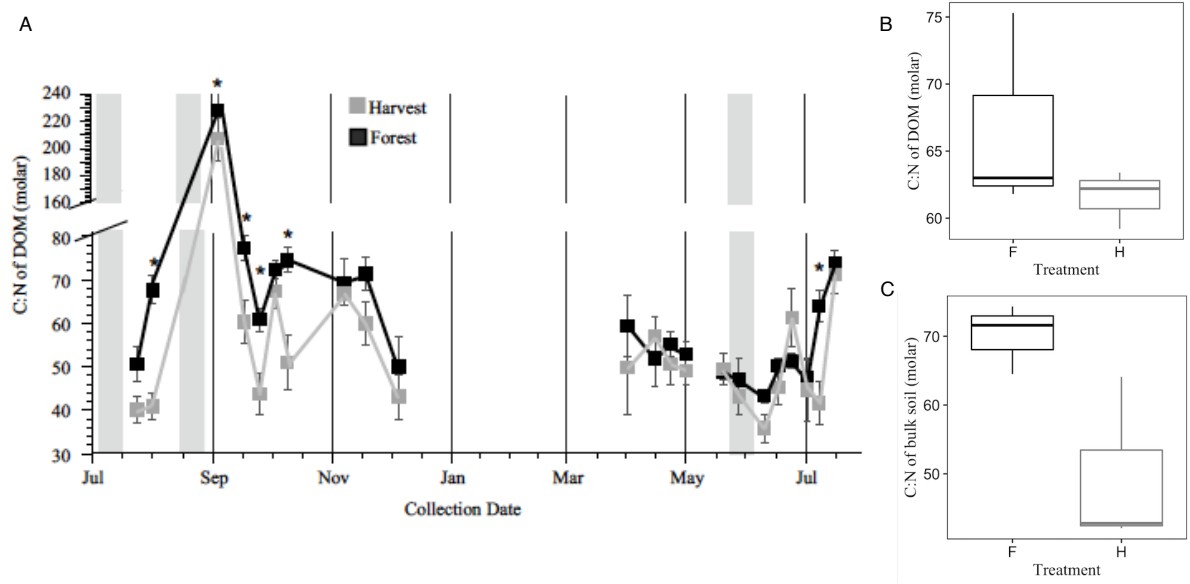


**Figure 2.** Intra-annual variation of the dissolved organic carbon (DOC) to dissolved organic nitrogen (DON) ratio collected by lysimeters (a) located in plots within the Pynn's Brook Experimental Forest in western Newfoundland, Canada. Each point is a mean of 12 lysimeters per treatment per collection day. Grey bars designate periods of 10 consecutive days receiving less than 10 mm/day of precipitation. Breaks in the line

graphs between points indicate periods of time when a sampling attempt was made but no water was captured by the lysimeters, indicating a zero flux. Asterisks indicate collection days where significant differences between treatments occurred as determined by a repeated measures ANOVA and post hoc least square means test, alpha = 0.05. Boxplot of C:N of DOM in harvest (H) and forest (F) treatments (b) compared to organic horizon C:N in H and F (c). Boxplots show the median (line inside box), upper and lower quartiles (box top and

bottom, respectively), and minimum and maximum values (error bars) associated with plot scale annual means (n=3 per plot type).




**Table 1.** Repeated measures linear mixed effects model results assessing the effect of collection day and the interaction with treatment on mobilized soil concentrations of total dissolved nitrogen (TDN), ammonium ($NH_4^+$), dissolved organic nitrogen (DON), orthophosphate ($PO_4^{3-}$), as well as the ratios between $NH_4^+$ and TDN and between dissolved organic carbon (DOC) and DON (DOC:DON) collected from plots located within the Pynn's Brook Experimental Forest in western Newfoundland, Canada.

| [TDN] | df | F-value | p-value |
|---|---|---|---|
| Treatment | 1 | 0.429 | 0.5193 |
| Day | 22 | 35.7732 | **<0.0001** |
| Treatment x Day | 22 | 2.347 | **0.0006** |
| [$NH_4^+$] | df | F-value | p-value |
| Treatment | 1 | 0.14539 | 0.7066 |
| Day | 22 | 12.58802 | **<0.0001** |
| Treatment x Day | 22 | 2.27061 | **0.0009** |
| [DON] | df | F-value | p-value |
| Treatment | 1 | 17.3581 | 0.0004 |
| Day | 21 | 35.5673 | **<0.0001** |
| Treatment x Day | 21 | 11.1212 | **<0.0001** |
| [$PO_4^{3-}$] | df | F-value | p-value |
| Treatment | 1 | 1.76 | 0.1983 |
| Day | 22 | 11.03 | **<0.0001** |
| Treatment x Day | 22 | 2.69 | **0.0001** |



**Table 2.** Pearson correlations between lysimeter captured dissolved organic matter (concentrations, ratios and fluxes) and environmental variables (soil temperature, moisture and water input) within mature forest (F) and harvested (H) treatments plots located within the Pynn's Brook Experimental Forest in western Newfoundland, Canada. Dissolved organic carbon (DOC), total dissolved nitrogen (TDN), dissolved organic nitrogen (DON), ammonium ($NH_4^+$), and orthophosphate ($PO_4^{3-}$). Bold font highlights significant correlations. A Bonferroni correction was applied to account for Type 1 Error (alpha = 0.05/3 = 0.017).

| | | df | A. soil temperature | | B. soil moisture | | C. water input | |
|---|---|---|---|---|---|---|---|---|
| | | | F | H | F | H | F | H |
| Concentrations | [DOC] mg C L⁻¹ | 23 | r= 0.9493 t= 7.7154 **p< 0.0001** | r= 0.8083 t= 6.5847 **p<0.0001** | r= -0.2383 t= -1.177 p= 0.2512 | r= -0.4773 t= -2.6052 **p= 0.01582** | r= -0.4325 t= -2.3008 p= 0.0308 | r= -0.5431 t= -3.1022 **p= 0.0050** |
| | [TDN] mg N L⁻¹ | 21 | r= 0.7708 t= 5.5451 **p<0.0001** | r= 0.7038 t= 4.5409 **p= 0.0001** | r= 0.0727 t= 0.3340 p=0.7412 | r= -0.1563 t= -0.7253 p= 0.8305 | r= -0.5010 t= -2.6529 **p= 0.0148** | r= -0.5398 t= -2.9388 **p= 0.0078** |
| | [DON] mg N L⁻¹ | 20 | r= 0.2390 t= 1.1001 p= 0.284 | r= 0.6749 t= 4.088 **p= 0.0006** | r= 0.2203 t= 1.010 p= 0.3245 | r= -0.0864 t= -0.3880 p= 0.7021 | r= -0.0978 t= -0.4398 p= 0.6648 | r= -0.5242 t= -2.7531 **p= 0.0123** |
| | [NH₄⁺] mg N L⁻¹ | 21 | r= 0.6835 t= 0.7038 **p= 0.0003** | r= 0.7357 t=4.2911 **p<0.0001** | r= 0.01914 t= 0.08775 p= 0.9309 | r= -0.3721 t= -1.8375 p= 0.0803 | r= -0.6081 t= -3.5102 **p= 0.0021** | r= -0.5170 t= -2.7677 **p= 0.0115** |
| | [PO₄³⁻] mg P L⁻¹ | 21 | r= 0.6309 t= 3.7269 **p= 0.0012** | r=0.6592 t= 4.017 **p=0.0006** | r= -0.4021 t= -2.0123 p= 0.0572 | r= -0.2287 t=-1.077 p=0.2939 | r= -0.3933 t= -1.9604 p= 0.0633 | r= -0.2904 t= -1.3912 p= 0.1787 |
| Ratios | DOC:DON | 20 | r= 0.3279 t= 1.55 p= 0.1362 | r= 0.1750 t= 0.79 p= 0.436 | r= -0.5644 t= -3.0578 **p= 0.00621** | r= -0.5079 t= -2.6372 **p= 0.0158** | r= -0.068 t= -0.3066 p= 0.7623 | r= -0.068 t=-0.3077 p= 0.7623 |
| | NH₄⁺:TDN | 20 | r= 0.4556 t= 2.2891 p= 0.0331 | r= 0.2945 t= 1.3781 p= 0.1834 | r= -0.2863 t= -1.3365 p= 0.1964 | r= -0.4676 t= -2.366 p= 0.0282 | r= -0.44 t= -2.194 p= 0.0402 | r= -0.30 t= -1.4273 p= 0.1689 |
| Fluxes | g DOC m⁻²d⁻¹ | 28 | r= -0.1387 t= -0.7412 p= 0.4647 | r= -0.1575 t= -0.8437 p= 0.4060 | r= -0.1282 t= -0.6843 p= 0.4994 | r= -0.1454 t= -0.7779 p= 0.4431 | r= 0.7358 t= 5.7500 **p<0.0001** | r= 0.6113 t= 4.0880 **p= 0.0003** |
| | g TDN m⁻²d⁻¹ | 26 | r= -0.3371 t= -1.8258 p= 0.0793 | r= -0.2691 t= -1.4252 p= 0.1660 | r= 0.01418 t= 0.0723 p= 0.9429 | r= -0.0802 t= -0.4102 p= 0.6850 | r= 0.8243 t= 7.1343 **p<0.0001** | r= 0.6610 t= 4.4925 **p= 0.0001** |
| | g DON m⁻²d⁻¹ | 24 | r= -0.3917 t= -2.0858 p= 0.0478 | r= -0.3127 t= -1.6130 p= 0.1198 | r= 0.0625 t= 0.3069 p= 0.7615 | r= -0.0134 t= -0.0659 p= 0.9480 | r = 0.7374 t= 5.2356 **p<0.0001** | r= 0.6627 t= 4.3356 **p= 0.0002** |
| | g NH₄⁺ m⁻²d⁻¹ | 26 | r= 0.0528 t= 0.2700 p= 0.7892 | r= 0.1340 t= 0.6899 p= 0.2251 | r= -0.0829 t= -0.4242 p= 0.6749 | r= -0.2031 t= -1.0582 p= 0.2997 | r= 0.5101 t= 3.0239 **p= 0.0056** | r= 0.3769 t= 2.0753 p= 0.0479 |
| | g PO₄³⁻ m⁻²d⁻¹ | 26 | r= 0.0232 t= 0.1187 p= 0.9064 | r= 0.2367 t= 1.2426 p= 0.4209 | r= -0.2663 t= -1.4090 p= 0.1707 | r= -0.1855 t= -0.9627 p= 0.3445 | r= 0.5771 t= 3.6033 **p= 0.0013** | r= 0.2916 t= 1.5545 p= 0.1322 |

negatively correlated with water input (Table 2c), except for $PO_4^{3-}$ in both treatments and $NH_4^+$ in H. All fluxes were positively correlated to the water input into the soil (Table 2c), except $NH_4^+$ and $PO_4^{3-}$ in H. No relationship was observed between concentrations and fluxes with soil moisture, except the negative correlation with DOC concentration in the H treatment (Table 2b). Soil moisture was negatively correlated with the C:N of DOM in both treatments.

**3.3 Seasonal fluxes and concentrations in harvested and forest treatments**

An effect of season ($p<0.0001$) and treatment ($p = 0.0358$) was observed on total soil water fluxes with no interactive effect (Table 3). Soil water fluxes were always greater through the O horizons of H treatment compared to F and the four seasonal periods (summer, autumn, winter and snowmelt/spring; see *Seasonal Designations* in methods) exhibited four different cumulative water fluxes (Fig. 3a). The largest cumulative water fluxes in H and F treatments occurred over the snowmelt period and the smallest water fluxes occurred during the winter when a consistent snowpack resulted in very low inputs of water to the soil. The second largest flux of water occurred during autumn, the only seasonal period when water fluxes were significantly different between treatments. A relatively small cumulative water flux occurred during the summer period, though still larger than the overwinter flux. All DOM fluxes exhibited an effect of season (Table 3; $p<0.0001$), but an effect of treatment was only observed for DON flux ($p = 0.0167$). No interactive effect of season and treatment on DOM fluxes was observed.

The largest total flux of DOC (Fig. 3b) occurred during the autumn, and intermediate fluxes of DOC occurred in the summer and during snowmelt, which were not significantly different. The smallest total flux of DOC occurred during the winter. The largest total fluxes of DON occurred during both autumn and snowmelt periods. An intermediate flux of DON occurred during the summer, and the smallest flux occurred during the winter (Fig. 3c). The relative seasonal DOC and DON patterns described above resulted in C:N of DOM that was highest in the summer, decreased in autumn, and was lowest during winter and snowmelt (Fig. 3d). The variation in values for the water, DOC and DON flux is likely a result of variation in litterfall and throughfall across the plots overall and observed through the individual lysimeters (Bowering et al., 2020).

There was an effect of collection date on all absorbance properties (Table 4a-d; $p<0.0001$), and an effect of treatment on $SUVA_{254nm}$ only where values were often higher in the H treatment ($p = 0.0033$). An interactive effect was observed for $SS_{275-295nm}$ ($p = 0.001$) and $SS_{350-395nm}$ ($p = 0.0045$). An effect of collection date on both Fe and Al (Table 4e,f; $p<0.0001$) was also observed but only Fe exhibited a treatment effect ($p = 0.0332$) with

higher concentrations in the H treatment.  There was an effect of collection day on DOC:Fe (Table 4g; $p<0.0001$) and no effect of harvesting while only a harvesting effect was observed with DOC:Al with higher values observed in the F relative to H treatment (Table 4h; $p=0.0247$). The elevated values and variability in DOC:Fe and DOC:Al as well as the lack of collection date effect for DOC:Al indicates little evidence for relevant seasonal controls on proportions of these reactive metal that might impact the fate of organic horizon

DOC. The optical properties of CDOM in snow, collected as a bulk snow core of the entire profile just prior to snowmelt, contrasted with that of the lysimeter samples (Table 5). The snow $SUVA_{254nm}$ values were lower than all lysimeter samples and the LMW spectral slope ($SS_{275-295nm}$) was higher than lysimeter samples collected during snowmelt. The HMW spectral slope ($SS_{350-395nm}$) of snow was higher in F than H treatments, and was higher in F snow samples than F lysimeter samples. The large differences in HMW spectral slope between

treatments, compared to the LMW spectral slope, resulted in an elevated $S_R$ value for the H (2.60) in comparison to the F (0.69) treatment snow. This is likely attributable to greater litterfall sources of DOM in the F treatment plots given the greater needle litterfall on the snow surface and higher input of DOC from snow in the F relative to H plots (2.1 versus 1.3 g DOC $m^{-2}$ $y^{-1}$ for F and H plots, respectively; Bowering et al., 2020).

A principal component analysis (PCA) including absorbance properties ($SUVA_{254nm}$, $SS_{275-295nm}$, $SS_{350-395nm}$, and

$S_R$), the C:N of DOM and DOC:Fe grouped by treatment (Fig. 4a) and season (Fig. 4b) demonstrated the overriding effect of season compared to harvesting. PC1 and PC2 describe 41% and 26.6% of the dataset variability, respectively. Seasonally, autumn soil DOM is characterized by HMW CDOM, signified by higher $SS_{350-395nm}$ and lower $S_R$, as well as higher C:N of DOM, while winter and snowmelt samples are characterized by LMW CDOM signified by higher $S_R$, and to a lesser extent higher $SUVA_{254nm}$ and lower C:N. The samples

from the H and F plots were weakly separated by $SUVA_{254nm}$ and DOC:Fe, with higher values of both in the H treatment.



**Table 3.** Repeated measures linear mixed effects model results assessing the effect of treatment, season and their interaction on the total O horizon flux of water, dissolved organic carbon (DOC), dissolved organic nitrogen (DON), and 430 the ratio between DOC/DON determined from plots located within the Pynn's Brook Experimental Forest in western Newfoundland, Canada. Seasonal variations of water flux, DOC flux, DON flux and DOC:DON are shown in Figure 3.

| A. Water flux | df | F-value | p-value |
|---|---|---|---|
| Treatment | 1 | 4.99823 | **0.0358** |
| Season | 3 | 33.37198 | **<0.0001** |
| Treatment x Season | 3 | 2.24488 | 0.0912 |
| B. DOC flux | df | F-value | p-value |
| Treatment | 1 | 1.51888 | 0.2308 |
| Season | 3 | 31.85004 | **<0.0001** |
| Treatment x Season | 3 | 2.52235 | 0.0653 |
| C. DON flux | df | F-value | p-value |
| Treatment | 1 | 6.70889 | **0.0167** |
| Season | 3 | 31.0272 | **<0.0001** |
| Treatment x Season | 3 | 1.97875 | 0.1257 |
| D. DOC:DON | df | F-value | p-value |
| Treatment | 1 | 1.3919 | 0.2507 |
| Season | 3 | 21.2403 | **<0.0001** |
| Treatment x Season | 3 | 0.4462 | 0.7208 |

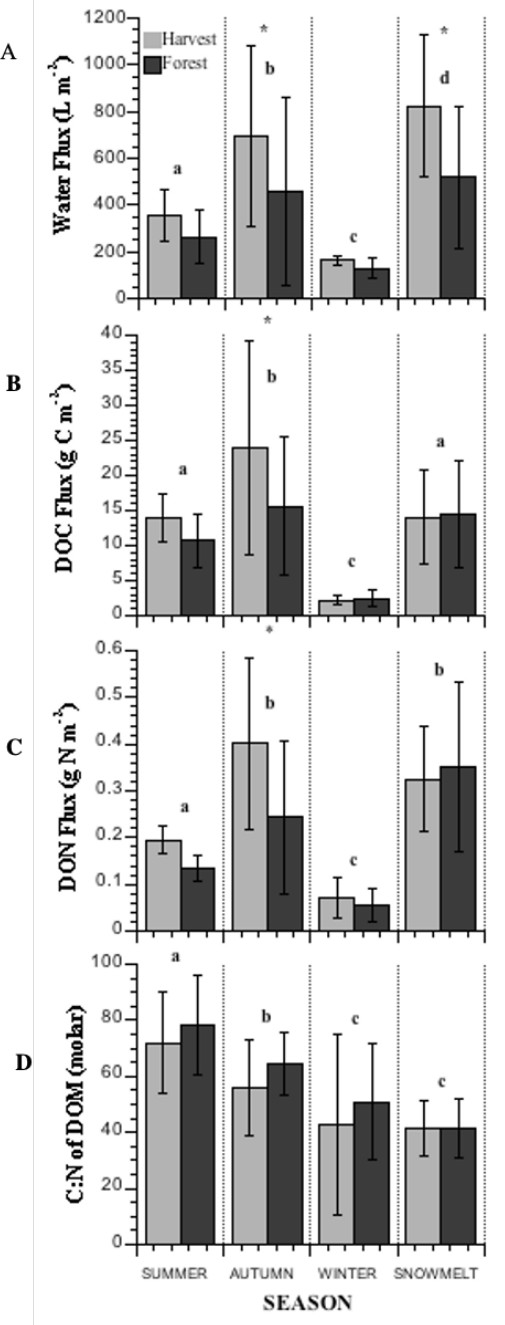

**Figure 3.** Total seasonal fluxes of water, dissolved organic carbon (DOC), dissolved organic nitrogen (DON) and the C:N of DOM in the mature forest and harvested treatment plots located within the Pynn's Brook Experimental Forest in western Newfoundland, Canada. Seasonal designations are described in the methods section. Seasonal periods sharing the same letter are not significantly different. Asterisks indicate significant differences between treatments. Error bars show the standard deviation of 12 lysimeters per treatment per season.

**Table 4.** Repeated measures linear mixed effects model results assessing the effect of treatment, collection day and their interaction on specific UV absorbance 254nm (SUVA), low molecular weight spectral slope (LMW), high molecular weight spectral slope (HMW), iron
concentrations ([Fe]), aluminum concentrations ([Al]), the ratio of dissolved organic carbon to iron (DOC:Fe) and the ratio of dissolved organic carbon to aluminum (DOC:Al) of mobilized soil dissolved organic matter collected from lysimeters across plots located within the
Pynn's Brook Experimental Forest in western Newfoundland, Canada.

| A. SUVA | DF | F-value | p-value |
|---|---|---|---|
| Treatment | 1 | 10.858 | **0.0033** |
| Date | 6 | 44.715 | **<0.0001** |
| Treatment:Date | 6 | 2.138 | 0.0539 |

| B. LMW | DF | F-value | p-value |
|---|---|---|---|
| Treatment | 1 | 1.67 | 0.2099 |
| Date | 6 | 18.81 | **<0.0001** |
| Treatment:Date | 6 | 5.09 | **0.0001** |

| C. HMW | DF | F-value | p-value |
|---|---|---|---|
| Treatment | 1 | 2.81 | 0.1077 |
| Date | 6 | 18.12 | **<0.0001** |
| Treatment:Date | 6 | 3.33 | **0.0045** |

| D. SR | DF | F-value | p-value |
|---|---|---|---|
| Treatment | 1 | 0.05 | 0.8238 |
| Date | 6 | 31.9 | **<0.0001** |
| Treatment:Date | 6 | 1.51 | 0.1857 |

| E. [Fe] | DF | F-value | p-value |
|---|---|---|---|
| Treatment | 1 | 5.16154 | **0.0332** |
| Date | 6 | 26.1303 | **<0.0001** |
| Treatment:Date | 6 | 1.34221 | 0.2439 |

| F. [Al] | DF | F-value | p-value |
|---|---|---|---|
| Treatment | 1 | 4.09043 | 0.0554 |
| Date | 6 | 19.13497 | **<0.0001** |
| Treatment:Date | 6 | 0.89863 | 0.4984 |

| G. DOC:Fe | DF | F-value | p-value |
|---|---|---|---|
| Treatment | 1 | 3.38677 | 0.0793 |
| Date | 6 | 6.81007 | **<0.0001** |
| Treatment:Date | 6 | 1.05731 | 0.3922 |

| H. DOC:Al | DF | F-value | p-value |
|---|---|---|---|
| Treatment | 1 | 5.81735 | **0.0247** |
| Date | 6 | 1.96941 | 0.0754 |
| Treatment:Date | 6 | 0.56551 | 0.757 |

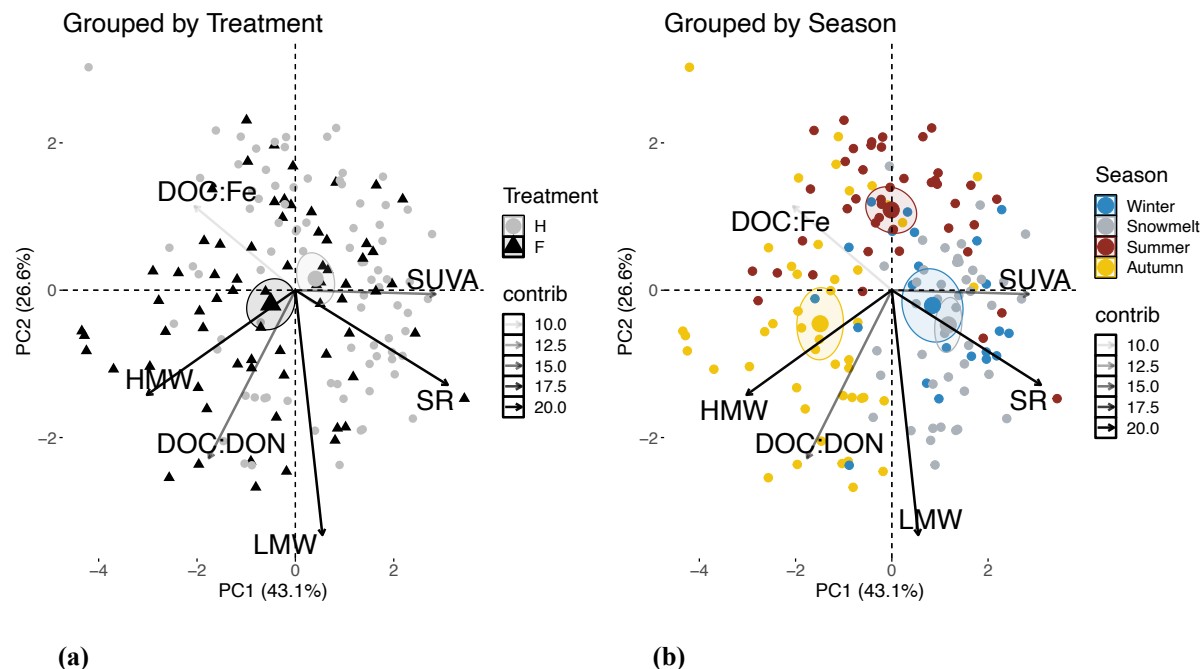

**(a)**                                                    **(b)**

**Figure 4.** Principal component analysis biplots used to explore the predominant variables describing the
harvesting effect compared to the seasonal effect on the composition of mobilized dissolved organic matter
(DOM) within the Pynn's Brook Experimental Forest in western Newfoundland, Canada. Variables included
are dissolved organic carbon (DOC) to iron ratio (DOC:Fe), the spectral slope for 350 to 395 nm (SS350-
395nm) indicative of high molecular weight (HMW) DOM, the spectral slope for 275 to 295 nm (SS275-
295nm) indicative of low molecular weight (LMW) DOM, the spectral slope ratio (SR), specific ultraviolet
absorbance at 254 nm (SUVA), and the C:N of DOM (DOC:DON). Treatments shown in (a) include samples
taken from the forest (F) and harvest (H) plots. Seasons shown in (b) include samples taken from Winter,
Snowmelt, Summer and Autumn. Vectors are shaded according to their combined percent contribution
(contrib) to PCA1 and PCA2. Ellipses represent the 95% confidence interval around each group mean.





**Table 5.** Optical properties and dissolved organic matter-metal associations within the Pynn's Brook Experimental Forest in western Newfoundland, Canada. Specific UV absorbance measured at 254nm (SUVA). Slope ratio (SR) is the spectral slope at 275–295 nm divided by the spectral slope at 350–400nm, pH, and Fe and Al concentration in mature forest (F) and harvested (H) treatments. Bolded values show significant treatment differences on certain sampling dates. Values with the same letter within each analysis type column are not significantly different. Standard deviations of the mean of 12 lysimeters per treatment shown in brackets. ND = No data.


| Date | DOC:DON (molar) | | pH | | SUVA L mg$^{-1}$ | | SR 275/350 | | [DOC] ppm | | [TDP] ppm | | [Fe] ppb | | [Al] ppb | | DOC:Fe (molar) | | DOC:Al (molar) | |
|---|---|---|---|---|---|---|---|---|---|---|---|---|---|---|---|---|---|---|---|---|
| | F | H | F | H | F | H | F | H | F | H | F | H | F | H | F | H | F | H | F | H |
| 17-Sep-13 | **77.39** | **60.35** | 3.65 | 3.87 | 3.64 | 4.12 | 0.74 | 0.73 | 71.09 | 60.94 | 0.37 | 0.32 | 178 | 258 | 569 | 707 | 2199 | 1606 | 461 | 259 |
| | (9.97) | (16.67) | (0.15) | (0.15) | (0.22) | (0.18) | (0.03) | (0.04) | (22.10) | (20.44) | (0.28) | (0.56) | (73) | (164) | (436) | (335) | (1178) | (1608) | (412) | (207) |
| 07-Nov-13 | 69.38 | 66.80 | 4.07 | 4.12 | **4.47** | **4.86** | 0.76 | 0.76 | 30.35 | 28.85 | 0.26 | 0.07 | **126** | **210** | 366 | 563 | 1374 | 815 | 302 | 169 |
| | (18.27) | (9.64) | (0.19) | (0.13) | **(0.19)** | **(0.32)** | (0.02) | (0.04) | (7.90) | (8.00) | (0.18) | (0.05) | **(53)** | **(126)** | (237) | (313) | (843) | (410) | (257) | (153) |
| 15-Apr-14 | 51.93 | 57.05 | ND | ND | 4.65 | 4.89 | 0.81 | 0.83 | 28.38 | 15.18 | 0.19 | 0.04 | 83 | 108 | 255 | 305 | **2091** | **806** | **435** | **141** |
| | (22.16) | (15.05) | | | (0.32) | (0.65) | (0.04) | (0.03) | (15.62) | (5.07) | (0.17) | (0.03) | (49) | (61) | (207) | (148) | **(1297)** | **(380)** | **(336)** | **(99)** |
| 01-May-14 | 53.01 | 49.17 | 4.30 | 4.41 | 5.06 | 5.34 | 0.79 | 0.80 | 15.19 | 13.52 | 0.09 | 0.03 | 74 | 122 | 191 | 371 | 1417 | 667 | **404** | **127** |
| | (10.16) | (11.67) | (0.17) | (0.13) | (0.25) | (0.51) | (0.02) | (0.01) | (4.68) | (4.88) | (0.07) | (0.03) | (55) | (70) | (162) | (172) | (1013) | (340) | **(439)** | **(142)** |
| 20-May-14 | 48.86 | 49.47 | 4.99 | 5.13 | **5.12** | **5.47** | 0.082 | 0.83 | 11.21 | 13.48 | 0.05 | 0.04 | 54 | 107 | **146** | **359** | 1302 | 632 | **354** | **107** |
| | (6.08) | (12.14) | (0.31) | (0.15) | **(0.20)** | **(0.28)** | (0.02) | (0.01) | (2.03) | (5.02) | (0.02) | (0.06) | (40) | (54) | **(130)** | **(146)** | (676) | (280) | **(297)** | **(72)** |
| 10-Jun-14 | 43.29 | 35.89 | 4.84 | 4.84 | 4.95 | 4.87 | 0.78 | 0.76 | 45.08 | 31.91 | 0.23 | 0.29 | 137 | 171 | 393 | 470 | 1800 | 1072 | 400 | 187 |
| | (5.36) | (11.19) | (0.54) | (0.20) | (0.54) | (0.62) | (0.05) | (0.04) | (15.50) | (10.57) | (0.17) | (0.37) | (58) | (105) | (269) | (191) | (1021) | (486) | (319) | (110) |
| 08-July-14 | **64.07** | **41.63** | ND | ND | **4.06** | **4.64** | 0.74 | 0.76 | 62.27 | 37.17 | 0.23 | 0.16 | 156 | 185 | 439 | 484 | **2118** | **1091** | 461 | 192 |
| | **(12.40)** | **(16.95)** | | | **(0.19)** | **(0.46)** | (0.023) | (0.034) | (19.08) | (11.77) | (0.12) | (0.12) | (55) | (99) | (286) | (163) | **(1049)** | **(409)** | (305) | (86) |
| Snow 02-Apr-14 | ND | ND | ND | ND | 2.17 | 1.51 | 0.69 | 2.60 | 7.2 | 3.4 | ND | ND | ND | ND | ND | ND | ND | ND | ND | ND |

## 4 Discussion

This study provides evidence for a strong control of season on the chemical composition of dissolved organic matter (DOM) mobilized from boreal forest organic (O) horizons that supersedes controls attributed to the decadal time scale effect of forest harvesting. Clear-cut harvesting immediately reduces the interception of water through the removal of trees but also through the longer-term reduction of the O horizon thickness and associated moss layer. Similar to DOC fluxes (Bowering et al., 2020), clear-cutting increased the mobilization

of DON on weekly to annual time scales. The relative temporal patterns of DOM composition, however, were similar in the F (mature forest) and H (harvested) treatments supporting other work which describe a dominating

control of environmental factors compared to soil composition on DOM composition (Cronan and Aiken, 1985; Kaiser et al., 2001; Fröberg et al., 2011). The compositional patterns observed in this study are indicative of DOM from fresh plant and microbial origin in the summer through autumn, and a shift to DOM from microbial

biomass and microbially processed materials underneath a consistent snowpack in winter and during snowmelt. These seasonal shifts highlight a potential sensitivity of DOM composition to the changing climate, particularly in northern forests experiencing snowpack reductions and an increased autumn-winter wet period. The more dynamic snowpacks (those experiencing melt) associated with warmer winters can increase winter soil water fluxes, and may consequently drive the mobilization of less processed soil DOM (i.e. higher HMW and C:N)

during winter. This hypothesized soil response to increasingly warmer winters in this region is consistent with observed increases in river DOM export attributed to enhanced wintertime exports from a large catchment in a similar landscape further south (Huntington et al. 2016). The temporal and compositional changes observed here have implications for the ultimate lability, physiochemical and biological, and fate of soil DOM within northern forests and once exported to aquatic systems.


**4.1 Summer soil DOM reflects decomposition of plant products and N mineralization**

The high C:N and HMW DOM mobilized in summer is explained by the dominance of the decomposition of fresh plant litter releasing water soluble organic C relative to organic N. Summer in this forested landscape is a

period of relatively lower precipitation, high soil temperature, and multi-day periods of soil drying followed by rewetting. Decomposition of litter resulting in the release of soluble materials at high soil temperatures (when moisture is not limiting), results in the release of soil C and uptake or immobilization of N (Kirschbaum, 1995; Conant et al., 2011; Hilli et al., 2008). While greater proportions of soil C are mineralized and released as $CO_2$ during this period, a byproduct of greater microbial activity is greater production of soluble C, resulting in the

high DOC concentrations and low pH often observed at high soil temperatures in laboratory extractions (Moore et al., 2008; Lee et al., 2018), and in situ (Kalbitz et al., 2007; Bowering et al., 2020). Additional concurrent processes known to affect DOM production at high seasonal temperatures are soil drying and rewetting cycles (Fierer and Schimel, 2002), and rhizodeposition (Weintraub et al., 2006; Heijden et al., 2008). These processes could contribute to mobilized DOC in summer, although the later would contribute LMW DOM instead of the

HMW DOM observed (Giesler et al., 2007).

While the above processes result in an increase in DOC in summer, a number of other concurrent processes result in the transformation and uptake of dissolved ON. Higher rates of N mineralization likely contributed to the larger ratio of dissolved inorganic N (DIN) relative to total dissolved N (TDN) observed (Fig. 1),

highlighting the likelihood of greater ON processing during summer. No detectable nitrate in soil leachates along with low pH of soil solution, suggests that nitrification in this system is limited (Ste-Marie and Paré, 1999). In addition, direct uptake of DON by vegetation during the growing season is possible in northern latitudes that are N deficient (Neff et al., 2003; Schimel and Bennett, 2008; Näsholm et al., 1998) with plants and microbes competing for LMW DON, such as amino acids and peptides (Farrell et al., 2014). Combined,

these processes limit the amount of ON available for mobilization during summer and contribute to the elevated C:N of the DOM observed during the summer period.

**4.2 Autumn soil DOM indicates a progressive reduction in soluble C but maintenance of organic N**

Following the relatively warm, dry summer period, the reduced temperature combined with increased plant inputs and decreased plant N demands in autumn lead to shifts in composition of mobilized DOM. Autumn, defined here as the period of continuous leaching of soil, constant soil moisture, and decreasing soil temperatures, resulted in initially high C:N of DOM, that decreased over the season. The sudden decrease in C:N of DOM observed at late autumn (Fig. 2), suggests that the O horizon had been leached of much of the

soluble organic C, while the available soluble organic N was maintained. Decomposition of litter and soil during summer in boreal coniferous forests is dominated by fungi, whose activity rely on seasonally dependent rhizodeposition (Žifčáková et al., 2017). Two important C inputs associated with photosynthesis are therefore likely reduced in this system in late autumn: that from rhizodeposition and that from rhizo-dependent fungal decomposition of litter. In contrast to organic C trends, continued rapid cycling of organic N has been observed

in northern black spruce forests of Alaska, even at low soil temperatures (Kielland et al., 2007) suggesting that continued breakdown of proteins replenishes the soluble ON pool during autumn. Furthermore, if DON uptake by plants is a relevant mechanism in this system, as is true in other northern systems (Schimel and Bennett, 2008), the demand for ON would decrease as plant activity slows in late autumn, reducing competition between the plant and microbial community for ON. This, in addition to decreasing rates of N mineralization with

decreasing soil temperature contributes to the maintenance of the soluble ON pool compared to a decreasing soluble OC pool during the wet fall-to-winter transition.

**4.3 Winter and snowmelt soil DOM reflect soil microbial contributions underneath the snowpack**

Fluxes of low C:N, LMW DOM occurring during winter and the following snowmelt period were likely the result of reduced plant inputs and maintenance of soil microbial activity underneath the snowpack. The winter period in this study year was characterized by a thick, consistent snowpack, that maintained constant soil temperatures at 2°C in both treatments. The snowpack developed before decreasing ambient temperatures could freeze the soil, allowing conditions for significant microbial activity underneath a consistently deep snowpack (>40 cm; Brooks et al., 2011). Soils under shallower snowpacks are more vulnerable to freeze-thaw events, resulting in fluctuations in microbial biomass through winter and periodic release of labile C (Schimel and Clein, 1996; Patel et al., 2018), such as carbohydrates and amino sugars (Kaiser et al., 2001). This can have significant impacts on growing season soil and stream DOC (Haei et al., 2010) and DON (Groffman et al., 2018) concentrations. In the absence of freeze-thaw cycles, cell lysis events may not be a significant mechanism of DOM release. Instead, microbial activity and decomposition of soil organic matter underneath the snowpack is likely the dominant source of DOM production over the duration of the winter. The low C:N is indicative of a greater microbial contribution to the DOM during this period. Interestingly, the winter and snowmelt samples exhibited high $SUVA_{254nm}$ coupled with elevated $S_R$ values, suggesting the mobilization of relatively more aromatic-rich DOM but with a lower molecular weight in comparison to soil DOM from summer and autumn. $SUVA_{254nm}$ and $S_R$ are typically negatively correlated in surface water indicative of HMW aromatic DOM (Helms et al. 2008). These results support the occurrence of microbial degradation of soil organic matter in the absence of fresh litter inputs underneath the snowpack, increasing the solubility of aromatic compounds such as lignin in soil (Malcolm, 1990; Hansson et al., 2010; Klotzbücher et al., 2013) enhancing aromaticity and reducing the relative molecular weight of the CDOM. Congruent with this, the lower DOC:DON is attributable to microbial metabolites or proteinaceous products released following this long period of degradation in absence of fresh plant inputs as observed in soil incubations (Kalbitz, 2003). These observations are also consistent with the initial steps in the synthesis of CDOM or humic substances; the fungal breakdown of lignin-cellulose as part of the polyphenol theory of humic substance formation processes (Stevenson 1994) also observed in organic rich peat (Prijac et al., 2022).

**4.4 The seasonal variability of soil DOM composition has implications for its fate in a changing climate**

This study demonstrates that the composition of mobilized soil DOM is similarly variable between sites of contrasting forest stand and soil properties. Clear-cut harvesting causes changes to forest water balance, through immediate removal of the canopy and longer-term reduction of the O horizon thickness, and results in larger quantities of mobilized DOM on decadal time scales. Despite this, the response of soil DOM composition to season suggests that the mobilization of soluble materials in the two treatments are controlled by similar responses to environmental conditions. Optical properties and the C:N of DOM during summer and autumn, compared to winter and snowmelt were reflective of shifts from fresh plant-derived to microbial-derived DOM. This compositional shift is especially noteworthy because temperature change at high-latitudes is expected to be more pronounced in the winter, with repercussions on snowpack formation and duration (Mellander et al., 2007; Laudon et al., 2013). Winter temperatures in this northern maritime climate are often near 0°C, and the projected 7°C increase in mean winter temperature (Finnis and Daraio, 2018) is likely to cause increases in winter rain and melt events, soil water fluxes and, consequently, mobilization of less processed soil DOM. The delivery of less processed DOM, exhibiting a higher MW CDOM fraction, could result in increased mineral soil adsorption (Guggenberger and Zech, 2002; Kaiser and Guggenberger, 2000; Lilienfein et al., 2004; Kothawala et al., 2008) dependent on the texture and mineralogy of the underlying soil and including the pathway and resident time of water (e.g. Oades, 1988; Marschner and Kalbitz, 2003; Patrick et al., 2022; Slessarev et al., 2022).

Research conducted in these same and similar forest sites underlain by podzolic soils indicate adsorption of DOM within shallow mineral soils is highly dynamic and dependent upon seasonal conditions (e.g. solution pH), and reactive Al availability and its saturation with C. The siliciclastic sedimentary till in the study area provides an ample source of reactive Fe and Al to support the generation of organo-mineral complexes via co-precipitation of DOM and these metals as well as adsorption onto existing complexes (Patrick et al. 2022). In some hillslope locations of this study site, deeper mineral horizons have a large potential for DOM adsorption dependent on infiltration depths that may occur with the enhanced fall and winter rainfall expected in this region (Patrick et al., unpubl. data). Therefore, the differences in DOM composition between fall and spring snowmelt further support potential for enhanced formation of these complexes with the expected decreases in spring snow melt and increasing autumn and winter rainfall and infiltration.

Warmer winters are linked to enhanced catchment export of DOM in an area just south of our study region (Huntington et al. 2016), consistent with regional evidence for enhanced soil DOM mobilization with warming winters (Bowering et al. 2022), and suggests a shift to increased export of fresher terrestrial DOM. In the freshwater environment fresher terrestrial DOM, characterized by an increased high MW fraction, degrades more readily relative to low MW, N-rich DOM (Kellerman et al., 2015; Köhler et al., 2013; Kothawala et al., 2014). However, fresher terrestrial DOM also contributes more to sediment burial in high latitude lakes through light and microbial mediated flocculation (Wachenfeldt et al., 2008, 2009), and can have a greater potential to remain buried longer relative to more microbially derived OM (Gudasz et al., 2012). Enhanced export of more labile, higher MW DOM coupled with enhanced water input, and thus lower water residence times in the aquatic environment such as in this study region, suggests increases in a high MW DOM fraction throughout the aquatic environment (Weyhenmeyer et al., 2014) and potential for lake burial or export to the marine environment opposite of those regions experiencing decreases in precipitation (Catalán et al., 2016). Therefore, the ultimate impact of these shifts in forest soil DOM composition on the C balance of these landscapes depends on both the connectivity with and properties of deep mineral soils as well as the net effect on downstream aquatic processes of burial and metabolism.

**5 Conclusion**

Future reductions in snowpack depth and duration as a result of increasing air temperature has the capacity to disrupt an important period of soil organic matter processing by microbes with repercussions on the composition of mobilized DOM, however these effects are dependent on the type of snowpack change (Stark et al., 2020). In this wet eastern boreal region, snowpacks are deep (>80 cm maximum snow depth) and are likely to change with warmer winters, increases in rain on snow events, and ice incasement, with soil freezing being a less significant concern in comparison to forests with shallower snowpacks. Future work capturing variable snowpack years (either within or across sites) and impacts on landscape and small catchment hydrology is needed. This would help clarify the relative importance of these changes on the chemical character of soil organic matter and mobilized soil DOM, and inform their implications for the fate of DOM within deeper mineral soils and the aquatic environment. Such efforts would improve our understanding of the response of soil organic matter to a rapidly changing climate and our ability to manage forest C balance in boreal landscapes.

**Data availability**

All data associated with this study can be found at this publically available site:
https://github.com/ArkosicGreywacke/PBEWA/tree/main/Hillslope/Pynn's%20Brook%20Experimental%20Forest%20Lysimeters%202013-2014

**Author Contribution**

KAE and SEZ designed the study with input from KLB. KLB and KAE designed the lysimeters and planned their installation as well as installation of all environmental monitoring equipment. KLB collected and analysed the lysimeter, environmental monitoring and soil properties data. KLB prepared the paper with significant input from SEZ and editing from KAE and SEZ.

**Competing interests.** The authors declare that they have no conflict of interest.

**Acknowledgements**

Special thanks for field assistance provided by individuals at the Atlantic Forestry Centre (Corner Brook) of Natural Resources Canada: Andrea Skinner, Darrell Harris, and Gordon Butt; and Memorial University, Grenfell campus: Sarah Thompson and Danny Pink, as well for laboratory assistance provided by Jamie Warren at Memorial University, St. John's campus.  Financial support for this research came from the Natural Sciences and Engineering Research Council (NSERC) Discovery Grants and Strategic Partnerships programs (grant no. 479224-15), the Canada Research Chairs Program, and the Centre for Forest Science and Innovation (Agrifoods and Forestry, Government of Newfoundland and Labrador).

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
