# Peer review of "Seasonal controls override forest harvesting effects on the composition of dissolved organic matter mobilized from boreal forest soil organic horizons"

_EGUsphere, 2022_

## Author Response (AR1)

Dear Dr. Buscardo:

Thank you for your handling of our manuscript and the constructive feedback from yourself and both reviewers. We have fully addressed all of the reviewers' comments including a full description of the study site (e.g. above-ground vegetation, age of regenerated trees, soil type, geological parent material) as well as inclusion of a more explicit linkage between climate change and DOM composition in relationship to the soil type and geological parent material. This later aspect was addressed to some extent in the abstract but in more detail within the discussion in the revised paper.

Please find our detailed responses to the reviewers' comments including approximate line number within the edited version with tracked changes below in *italics* after each comment.

Again, thank you for your input and handling of this paper. We look forward to your further input and decision.

Sincerely on behalf of all authors,

Sue Ziegler

Dear Dr. Bowering

I have read your rebuttal letter in response to the comments of the two reviewers and I am happy for you to proceed and post the revised version. Please provide a full description the of the study site (including details on above-ground vegetation composition / structure and age of naturally regenerated trees at the harvested plots, soil type / geology and climate) to guarantee the reproducibility and the interpretability of your results. Please discuss (and do not limit it to the abstract) the link between climate change and DOM composition in relation to soil type / geology at your study site.

Thank you.

Sincerely,

Erika Buscardo

**Comment on egusphere-2022-1418**

Anonymous Referee #1

Referee comment on "Seasonal controls override forest harvesting effects on the composition of dissolved organic matter mobilized from boreal forest soil organic horizons" by Keri L. Bowering et al., EGUsphere, https://doi.org/10.5194/egusphere-2022-1418-RC1, 2023

This study investigates the seasonal variations in dissolved organic matter fluxes and concentrations from the organic layer of boreal forests. Sites that have been logged ten years ago are compared with unharvested controls in a replicated experimental design.

The study design is sound, and the information generated is original and pertinent. DOM originating from the organic layer of boreal forests represent an important and poorly known flux of carbon that may be affected by disturbances and climate change. The factors that control this flux as well as those that control the quality of DOM are poorly known. This study brings new knowledge on how disturbances and climate change may affect DOM originating from boreal forest organic horizons. The results, showing a similar response of DOM to climate for both harvested and control plots as well as an important effect of the season are pertinent to better understanding the C cycling of boreal forests.

*Response: Thank you for your positive feedback and constructive comments.*

Comments:

To better understand the results, more information is needed concerning the experimental design. Specifically, what is the status of regeneration on the harvested plots? Is it bare soil? What is the status of tree regeneration, 10 years after harvesting? Has the site been recolonized by trees? What is the vegetation height, % cover... This information would be useful to understand the results. It is interesting that the differences in water and DOC fluxes are still present 10 years after harvesting. Another study conducted in somewhat similar conditions found coherent results with this one. However, it was conducted one year after harvest: warmer and wetter conditions in harvested treatment led to greater soil humidity, higher soil temperatures, and greater ammonification and DON production of harvested plots in the fall season (Coulombe et al. 2016). Can the author comment on the reason for a 10-year effect?

*Response: Thank you for raising these questions. We have provided further details in section 2.1 Site Description (Lines122-129; 135-146) to include these important details that will help provide critical context. The harvesting practices left the soil intact, however, post-harvest planting did not take place (the normal practice in the region). Therefore, regeneration of conifers remained sparse at the time of this study with the few trees that were established below 1.5 m in height at the time of the study. Soil surfaces of these harvested plots also had naturally recovered moss, herb and shrubbery at the time of this study. Part of this can be visualized in figure S1, an aerial photo taken not long after the initiation of the study (Figure S1).*

*We did expect 10-year post harvest effects attributed to a reduction in both litter and moss inputs, increase water input and decrease water residence times in surface soils of the harvested plots, and the consequences of these impacts on the organic layer physical and chemical*

*composition (e.g. decreased thickness; Bowering et al. 2020). We have added information on these differences in this section as well to help clarify relevant treatment differences and potential for their impacts on soil DOM composition (Lines 146-148).*

Minor comments

l.54: fate of DOM: there is abundant literature on the importance of mineral soil properties on the fate of DON that could be mentioned here.

*Response: Good idea. We feel this is best incorporated where we discuss the importance for understanding the composition of the surface sources of DOM studied here because this study does not focus on organo-mineral interactions. We have inserted this information within later lines in the introduction as:*

While soil extractions provide valuable information on potential sources, bioavailability, and production mechanisms of soil DOM (i.e. (Jones and Kielland, 2012)(Hensgens et al., 2020), as well as transformation and fate in mineral soil (Kothawala et al., 2008), they cannot capture the interaction of these factors with local hydrometeorological conditions important to understanding the net movement of DOM *in situ*.

l.74: recalcitrance of coniferous litter, how about that of bryophytes, are they not present?

*Response: Yes, forest floor mosses are quite abundant and important in these forests. We have inserted that information here with references for slow turnover of moss tissues.*

l.104: 10 years may not be considered long-term in a rotation that lasts more than 50 years; preferably use 10 years, – also line 535 and elsewhere.

*Response: This is a good point. We have removed "long-term" in both instances and clarified the 10-year time frame.*

l.381; l. 535 and elsewhere: the use of Sr for slope could be misleading (Strontium), use another notation.

*Response: We have changed this to $S_R$ throughout.*

Discussion/conclusions

Winter rain/thaw events are increasing. The results of this study suggest that these events may bring more transformed DOM into streams. Is it possible to support this claim by referring to studies in warmer ecosystems, such as coastal Maine or New England?

*Response: We agree it would be helpful to refer to catchment studies in the region or further south. However, with the predicted warmer winters our results suggest we would expect more organic soil layer derived DOM, but in a less transformed state, mobilized from these landscapes (see lines 623-635). Given this comment it looks like we need to clarify that, and will include reference to* Huntington et al., (2016) *where increasing trends in terrestrial inputs and riverine DOM to the Gulf of Maine (Penobscot River) associated with winter transport has been documented and thus consistent with our findings. Please see Lines 576-579.*

Reference:

https://doi.org/10.1139/cjfr-2016-0301 Coulombe et al. 2016. Effect of harvest gap formation and thinning on soil nitrogen cycling at the boreal–temperate interface

**Anonymous Referee #2**

Referee comment on "Seasonal controls override forest harvesting effects on the composition of dissolved organic matter mobilized from boreal forest soil organic horizons" by Keri L. Bowering et al., EGUsphere, https://doi.org/10.5194/egusphere-2022-1418-RC2, 2023

This study follows up on an earlier case study evaluating fluxes of DOC from harvested (10 years post harvest) and unharvested black spruce forest plots (n=4), in evaluating the composition of the dissolved organic matter through the season (n=3). A suite of solutes and DOM composition indices (absorbance spectrometry) were analyzed through four seasons. The harvest treatment effects were quite muted in comparison to seasonal changes, which is perhaps surprising given the purportedly different litter inputs between a 10 year-old aggrading forest and a mature forest. There was higher DON and DOC exported from organic horizons in the harvest treatment, but the effect size of this was small compared with seasonal changes. In agreement with other work, this work also demonstrated the diagenesis of DOM in surface soils through the winter, owing to snow buffering soil temperatures. The conclusions are, for the most part, supported by the work and the writing is clear and should be of broad interest. I offer minor comments by line number, and hope they are helpful in publishing this work.

*Response: Thank you for your constructive feedback. We found it instructive in improving this paper.*

Line 35 (abstract): How does this study inform on climate change effects?

*Response: We agree that this later portion of the abstract needed to be more direct and explicitly state how the work informs on climate change impacts. We have edited the last part of the abstract (see Lines 34-40) to indicate that the results (i.e. seasonal differences in O horizon DOM composition observed here, particularly between winter/spring melt and fall periods) indicate the trend of warmer winters and increased fall precipitation observed and anticipated with climate change has the potential to enhance delivery of DOM that is likely more reactive with mineral soils or photo- and bioreactive in the downstream freshwater environment. However, the impacts of this shift (soil OM sequestration or delivery to aquatic) will depend upon hydrology and specifically infiltration relative to lateral flow in these boreal landscapes.*

Line 90-95 (and throughout): There is a bit of text explaining the consequences of DOM quality from the O horizons for stabilization in the mineral soil, which seems like conjecture for the present study, which did not characterize organo-metallic complexation or stabilization in the mineral soil. I recommend tempering this. Moreover, soil type (for example, Podzols vs. Cambisols) is very important in mediating these processes.

*Response: We only make mention of the potential consequences of O horizon derived DOM composition within the mineral soil as a means to explain one important potential for the changes in composition. For example, in this specific case (Line 90-95) DOM with a higher*

*aromaticity is more likely to form organo-mineral complexes with reactive metals within the mineral soil environment. We agree, in this study we are not investigating fate of DOM in mineral soils nor organo-mineral processes and what controls them. Therefore, we have reviewed the paper and made revisions necessary to be certain that this is clearer (e.g. see Lines 110-112).*

Line 125: Some detail on the type of soil (taxonomy would be great) is needed. Even mean depths of L, F, H... type of parent material would be helpful

*Response: Thanks for raising this. We agree that this information is helpful and will add the soil type/classification (humo-ferro podzols; Canadian Soil Classification Work Group), mean depths of the O horizon (8.2 cm forested plots; 4.3 cm harvested plots; Bowering et al. 2020), and the parent material (till composition; generally less than 3 m, poorly sorted, with clasts of granites, porphyry, sandstone and siltstone (Batterson and Catto, 2003) with quartz most abundant, followed by plagioclase, muscovite, chlorite, K-feldspar and biotite (Patrick et al. 2023) for the study site and plots. This has been added to the site description section.*

Line 144: I thought there were four plots?

*Response: Indeed there are four plots of each treatment at the study site. However, in this study two pairs of lysimeters (four lysimeters) located in each of three forested and three harvested plots were used (n=12 per treatment or n=24 total). We have revised the study site and lysimeter sampling description to clarify this. Also note here that this design is illustrated in supplementary figure S1 which is now referred to more explicitly in that section to clarify.*

Line 170: Here, I was hoping to see some detail on typical snowpack for the region.

*Response: We should have included the information on regional climate (30-year normals) including snowfall within the site description section and have included that within the site description section. We also now provide the range of values for the study period itself such as the 1402 mm of total precipitation with 516 mm (37%) of that as snowfall in water equivalents within a region where snow typically represents ~40% of total precipitation.*

Line 196: What's a typical peak snow water equivalent at the freshet?

*Response: The snow water equivalent of the snowpack at the start of snowmelt period during the study was 83-110 mm as measured within the study plots, and was higher in the harvested plots (Bowering et al. 2020). We will include those values in the study site description. We know from the analysis of hydrographic data from three nearby stations, monitored by Environment Climate Change Canada (ECCC), and our own smaller headwater catchment sites, that ice jamming during this period precludes accurate estimation of the total freshwater discharge without use of conductivity and geomorphic information not available for the ECCC station data where some records for the region go back to 1963. Therefore, we feel it would be most informative to provide the snow on the ground water equivalent for the period of study to provide this information.*

Line 213: It is too bad that a conservative ion wasn't measured to act as a "tracer" (like Cl, or Br), as flux is likely to go down in H, yet solute concentrations are likely to go up, detracting from nailing down the mechanism for any changes (or not) in export in response to harvest treatment that account for concentration or dilution effects.

*Response: Indeed, utilizing a tracer in this study would have been helpful had we had access to precipitation and throughfall samples to define the tracer inputs which would vary significantly with time particularly in this maritime region.*

Line 362: It is interesting that summer fluxes were so high- as high as fluxes during the freshet!

*Response: Yes, we agree and our previous study suggests production limitation of the DOC flux over both the wet autumn and snowmelt periods (in contrast with summer when highest DOC concentrations are observed) likely controls the magnitude of DOC fluxes seasonally. This is important as it suggests that the impact of the current and project increasing trend in precipitation in the region on DOC mobilization from these landscapes will depend on the type and intra-annual distribution of that increased precipitation (Bowering et al. 2020).*

Line 378: Interesting. Why would the snow be different by treatment? Is this owing to differences in throughfall?

*Response: Yes, given the evidence for greater DOC concentration and HMW CDOM within the snowpack of forest plots relative to the snow collected from the harvested plots we suspect a greater source of litterfall derived DOM. For example, we observe greater needle litterfall on the snow surface and higher input of DOC from snow in the forest plots (2.1 versus 1.3 g DOC $m^{-2} y^{-1}$, forest and harvest, respectively). We will add a phrase in that section to help convey that point, briefly, citing the snow DOC contents reported by plot type in Bowering et al. (2020). See Lines 426-428.*

Figure 3: Do you think part of the autumnal differences could be owing to changes in litterfall between the harvest treatments?

*Response: Not exactly sure what differences are being referred to in this comment. Fig. 3 depicts treatment differences in water and DOC flux (elevated in harvested relative to forested) in autumn, and it also demonstrates that overall autumn fluxes of DOC and water are greater than in summer or winter, and similar to that observed over the snowmelt period. In the first instance (treatment effect) we hypothesize that the harvested plots exhibit a greater flux of DOC due to the enhanced water flux. In the second (seasonal), the combined effects of DOM source and water availability controls this flux, and explains the contrasts with the snowmelt attributed to the decomposition of soil OM under the insulated snowpack. However, this comment may also be referring to the large variation in values for the water, DOC and DON fluxes which certainly is likely capturing the variation in litterfall and throughfall across the plots overall as observed by the individual lysimeters. We have inserted a comment to direct the reader to this possibility (Lines 410-412).*

Line 346: What is the mechanism for "inconsistent snowpack" increasing soil water fluxes? Does this have to do with soil freezing, which has been shown to increase [DOC]? I'm assuming that this is meant to prime the reader, and you'll get to this later?

*Response: Thank you for raising the question as it is an important point to clarify. The reference to an inconsistent snowpack at the start of the discussion was meant to prompt the reader to consider how shifts in precipitation and form can result in changes to how much and what is mobilized by these soils. Here we are using "inconsistent" to refer to a dynamic snowpack, one that experiences melting, rather than just accrual, over the winter period; thus contributing to soil water fluxes during the winter period and less so to the larger spring melt period. In this region exposure of soil to freezing is not observed in these plots (at least not yet!). Our compositional results suggest such a shift, toward a more dynamic snowpack, could have consequences such as increasing mobilization of DOM that may have otherwise been decomposed over the course of the winter period under the insulation of snowpack (which generally prevents freezing in these soils as observed through soil T probes). We have edited this section to help clarify this (see lines 602-603).*

Line 565: Could you please provide a reference for rhizodeposited DOM having LMW?

*Response: Yes, we have added* Giesler et al., (2007) *to support this point.*

Line 610: This is indeed very interesting, and has been observed in peatlands as well (a negative relationship between SUVA254 and DOM oxidation, as well as DOM molecular weight). It could also be that relatively low molecular weight compounds are being polymerized (mainly by fungi). See for example, "polyphenol theory" described by F.J. Stevenson; "Humus Chemistry- genesis composition, reactions". 1994. pp: 188-211.

*Response: Thanks for this comment. We will briefly add reference to these congruent observations in peatlands and the fact that these observations are consistent with the initial steps in the theory for the generation of humic substances, namely the breakdown of lignocellulose (original plant polymers) into simpler lower molecular weight components that then contribute to reactions that form humic substances. Here we now refer to Stevenson (1994) and* (Prijac et al., 2022) *in this section.*

Line 638: I think it is also relevant to mention the importance of texture and parent material type for DOM adsorption and infiltration.

*Response: We now include more specific reference to the dependency of formation of organo-mineral complexes (OMCs) via DOM and metal co-precipitation and adsorption on to OMCs as a control on mineral soil adsorption of DOM as well as references (e.g. Oades, 1988; Slessarev et al., 2022; Patrick et al., 2022; Marschner and Kalbitz, 2003).*

*References cited in responses to both Reviewer 1 and 2:*

Batterson, M. J. and Catto, N. R.: Topographically-controlled Deglacial History of the Humber River Basin, Western Newfoundland, 1–16, 2003.

Giesler, R., Högberg, M. N., Strobel, B. W., Richter, A., Nordgren, A., and Högberg, P.: Production of dissolved organic carbon and low-molecular weight organic acids in soil solution driven by recent tree photosynthate, Biogeochemistry, vol. 84, 84, 1–12, https://doi.org/10.1007/s10533-007-9069-3, 2007.

Hensgens, G., Laudon, H., Peichl, M., Gil, I. A., Zhou, Q., and Berggren, M.: The role of the understory in litter DOC and nutrient leaching in boreal forests, Biogeochemistry, vol. 149, 149, 87–103, https://doi.org/10.1007/s10533-020-00668-5, 2020.

Huntington, T. G., Balch, W. M., Aiken, G. R., Sheffield, J., Luo, L., Roesler, C. S., and Camill, P.: Climate change and dissolved organic carbon export to the Gulf of Maine, Journal of Geophysical Research-Biogeosciences, vol. 121, 121, 2700–2716, https://doi.org/10.1002/2015jg003314, 2016.

Jones, D. L. and Kielland, K.: Amino acid, peptide and protein mineralization dynamics in a taiga forest soil, Soil Biology Biochem, vol. 55, 55, 60–69, https://doi.org/10.1016/j.soilbio.2012.06.005, 2012.

Marschner, B. and Kalbitz, K.: Controls of bioavailability and biodegradability of dissolved organic matter in soils, Geoderma, vol. 113, 113, 211–235, https://doi.org/10.1016/s0016-7061(02)00362-2, 2003.

Oades, J. M.: The retention of organic matter in soils, Biogeochemistry, vol. 5, 5, 35–70, https://doi.org/10.1007/bf02180317, 1988.

Patrick, M. E., Young, C. T., Zimmerman, A. R., and Ziegler, S. E.: Mineralogic controls are harbingers of hydrological controls on soil organic matter content in warmer boreal forests, Geoderma, vol. 425, 425, 116059, https://doi.org/10.1016/j.geoderma.2022.116059, 2022.

Prijac, A., Gandois, L., Jeanneau, L., Taillardat, P., and Garneau, M.: Dissolved organic matter concentration and composition discontinuity at the peat–pool interface in a boreal peatland, Biogeosciences, vol. 19, 19, 4571–4588, https://doi.org/10.5194/bg-19-4571-2022, 2022.

Slessarev, E. W., Chadwick, O. A., Sokol, N. W., Nuccio, E. E., and Pett-Ridge, J.: Rock weathering controls the potential for soil carbon storage at a continental scale, Biogeochemistry, vol. 157, 157, 1–13, https://doi.org/10.1007/s10533-021-00859-8, 2022.

---

## Author Response (AR2)

Dear Dr. Erika Buscardo:

Thank you for your handling of our manuscript including these further edits needed. We appreciate your catching these issues and have addressed them all, and provided the details on how they were addressed below after each listed in your original correspondence.

Sincerely,

Sue Ziegler on behalf of co-authors.

Dear Dr. Susan Ziegler,

Thank you for sending the revised version of your manuscript. I am satisfied with the way you addressed the reviewers' comments / suggestions both in the rebuttal letter and in the revised manuscript. There remain just few minor points that should be addressed.

Double-check all acronyms throughout the text. Sometimes they appear in full (e.g., carbon, forest, harvested) after being already defined, other times the acronym mentioned for the first time is not spelled out fully (e.g. SUVA, HMD, LMW, ON).

*Thank you for catching these issues. We have combed through the manuscript and made sure that these were all defined in their first use and then consisting used throughout. Exceptions to this include the abstract and figure captions standing alone from the main text.*

At the end of the Introduction substitute 'we frequently sampled' with a clear description of what was done, e.g., mobilised soil DOM was sampled on a weekly to monthly basis during a year... .

*We have added that needed detail so the sentence now reads "Mobilized soil DOM was sampled on a weekly to monthly basis over a year using a passive pan lysimeters in open harvested plots compared to adjacent mature black spruce forest plots as part of a case study"*

In the Material & Methods you list common plant species at the study site. In one case you partly identify the authority (i.e., Alnus alnobetula (Ehrh.) while for all the other species you opted for giving only the scientific name. Be consistent and either provide or not the authority for all species.

*Removed the one reference to authority to simplify and be consistent.*

Fig. and Figure / figure are used interchangeably throughout the text. Standardise.

Have converted all "Figure" to "Fig." throughout.

Avoid referring in the Material and Methods to tables and figures belonging to the Results.

*We removed the reference to Figs. 1, 2, 4a and 4b as well as Tables 3, 4, S1 and S2. from the Materials and Methods section. However, we retained the reference made to Fig. S1 and Fig. S2 as they provide site information that should help the reader understand the conditions as well as the experimental design, and do not contain any results.*

Figure legend / Table captions. They have to be self-standing, i.e. the reader needs to understand their content without reading the main text (e.g., the study site location is missing; the treatment is not defined in the table caption). Figure 2: space missing between 'harvest' and '(H)' and between 'forested' and '(F)'; Figure 3: no need for acronyms F and H.
Figures 1 and 2 – boxplots: describe correctly what the line, box and whiskers stand for.

*Done across all figures and tables including the errors noted. Thank you for catching all of those!*

Subtitle section 4.2: remove the full stop / period

*Done.*

Section 4.3, L632-35 - Revise sentence.

*This has been edited and now reads: "Interestingly, the winter and snowmelt samples exhibited high SUVA$_{254nm}$ coupled with elevated S$_R$ values, suggesting the mobilization of relatively more aromatic-rich DOM but with a lower molecular weight in comparison to soil DOM from summer and autumn"*

Thank you.

Sincerely,

Erika Buscardo

---

## Author Response (AR3)

Dear Erika

Thanks for all of the handling of our paper, has been very helpful in improving the manuscript.
Please find our revisions based upon the feedback/errors you provided.
We have added the boxplot details to Figure 2 and the study site location to Figure 4 and Table 5 captions.

Best,
Sue

Dear Dr. Susan Ziegler,

Thank you for sending the revised version of your manuscript. Your manuscript is now ready to be published. There are just a couple of minor things that need to be addressed.

Legend Figure 1: add boxplot details as done for Figure 2
Legend Figure 4 and caption Table 5: add study site location

Thank you again.

Sincerely,

Erika